# Vitamin K2 sensitizes the efficacy of venetoclax in acute myeloid leukemia by targeting the NOXA-MCL-1 pathway

**Tetsuzo Tauchi**[1]*, **Shota Moriya**[2], **Seiichi Okabe**[3], **Hiromi Kazama**[2], **Keisuke Miyazawa**[2], **Naoharu Takano**[2]*

1 Shinyurigaoka General Hospital, Asou-ku, Kawasaki, Kanagawa, Japan, 2 Department of Biochemistry, Tokyo Medical University, Shinjuku-ku, Tokyo, Japan, 3 Department of Hematology, Tokyo Medical University, Shinjuku-ku, Tokyo, Japan

* tauchi144@gmail.com (TT); ntakano@tokyo-med.ac.jp (NT)

**Data Availability Statement:** All relevant data are within the manuscript and its Supporting Information files.

## Abstract

Promising outcomes have been reported in elder patients with acute myeloid leukemia (AML) using combined therapy of venetoclax (VEN) and azacytidine (AZA) in recent years. However, approximately one-third of patients appear to be refractory to this therapy. Vitamin K2 (VK2) shows apoptosis-inducing activity in AML cells, and daily oral VK2 (menaquinone-4, Glakay[R]) has been approved for patients with osteoporosis in Japan. We observed a high response rate to AZA plus VEN therapy, with no 8-week mortality in the newly diagnosed AML patients consuming daily VK2 in our hospital. The median age of the patients was 75.9 years (range 66–84) with high-risk features. Patients received AZA 75 mg/m$^2$ on D1-7, VEN 400 mg on D1-28, and daily VK2 45 mg. The CR/CRi ratio was 94.7% (18/19), with a CR rate of 79%. Complete cytogenetic CR was achieved in 15 of 19 (79%) patients, and MRD negativity in 2 of 15 (13%) evaluable CR patients. Owing to the extremely high response rate in clinical settings, we further attempted to investigate the underlying mechanisms. The combination of VK2 and VEN synergistically induced apoptosis in all five AML cell lines tested. VK2, but not VEN, induced mitochondrial reactive oxygen species (ROS), leading to the transcriptional upregulation of NOXA, followed by MCL-1 repression. ROS scavengers repressed VK2 induced-NOXA expression and led to the cancellation of pronounced apoptosis and the downregulation of MCL-1 by VK2 plus VEN. Additionally, knockdown and knockout of NOXA resulted in abrogation of the MCL-1 repression as well as enhanced cytotoxicity by the two-drug combination, indicating that VK2 suppresses MCL-1 via ROS-mediated NOXA induction. These data suggest that the dual inhibition of BCL-2 by VEN and MCL-1 by VK2 is responsible for the remarkable clinical outcomes in our patients. Therefore, large-scale clinical trials are required.

## Introduction

BH3-mimetic agents that directly trigger apoptosis in cancer cells reliant on BCL-2 or its pro-survival relatives have emerged as powerful agents for treating chronic lymphocytic leukemia

**Funding:** This study was supported by funds provided through the JSPS KAKENHI (Grant Number 23K06658) to NT and JSPS KAKENHI (Grant Number 22K06653) to SM. All funder had no role in study design, data collection and analysis, the decision to publish or preparation of the manuscript. All funders provided consumable costs for this study.

**Competing interests:** The authors have declared that no competing interests exist.

(CLL) and acute myeloid leukemia (AML) [1]. Venetoclax (VEN) belongs to a novel BH3-mimetic class of small molecules that selectively target BCL-2, which binds to and suppresses pro-apoptotic proteins and activates the apoptosis effectors BAX and BAK to drive mitochondrial outer membrane permeabilization and cell death [1]. The current clinical data suggests that VEN, as a highly specific therapeutic agent, hits the bulk of CLL cells, which rapidly induces cell death by direct inhibition of BCL-2 because most CLL cells highly express BCL-2 and are reliant on BCL-2 [2, 3]. In contrast, in AML cells, there is heterogeneity in the response to VEN, reflecting the degree to which each cell is dependent on Bcl-2, and this varies between patients depending on the genetic and epigenetic features of each AML patient [4, 5]. For example, the anti-apoptotic protein MCL-1 is upregulated in VEN-resistant AML [6]. However, in AML cells, VEN also causes cell death in a different manner: direct inhibition of BCL-2 triggers a secondary wave that induces cell death [3, 7]. This may be due to the activation of other pro-apoptotic proteins through the dissociation of BCL-2 or a reduction in oxidative phosphorylation due to permeabilized mitochondria in AML cells [4, 7].

The use of VEN in clinical settings has resulted in promising outcomes in patients with AML [8]. Although VEN has shown limited activity as a single agent in patients with relapse, in recent studies, high response rates and promising remission periods were observed in older patients newly diagnosed with AML who were not candidates for intensive induction chemotherapy when treated with a methylation inhibitor or low-dose cytarabine [8, 9]. Treatment with VEN and azacytidine (AZA) is now recommended as the standard therapy for AML-unfit patients according to the NCCN clinical practice guidelines [10].

While we await confirmatory results, the high response rates reported for VEN-based combinations are exciting. However, up to one-third of patients appear to be refractory, and this population does not have obvious traditional biological risk factors, except for patients with TP53 mutations, who may have lower response rates [5, 11, 12]. In addition, the majority of newly diagnosed patients respond to VEN-based combination therapy, but the median duration of response is approximately one year, and we need to understand how resistance develops and how this can be targeted [11, 12].

We have previously shown that the vitamin K2 (VK2) analog menaquinone-4 (MK4) selectively and effectively induces apoptosis in various types of AML cells and blasts in myelodysplastic syndrome (MDS) [13–16]. An open-label single-arm prospective phase II clinical trial of VK2 in MDS, VK2 showed improvements in anemia and thrombocytopenia [17]. The effect of VK2 on normal hematopoietic progenitors also verified that VK2 induces normal myeloid progenitor differentiation and exhibits an anti-apoptotic effect on normal erythroid progenitors. [18]. MK4 (Glakay$^R$) has been approved to treat patients with osteoporosis due to its activity of γ-carboxylation of osteocalcin in Japan [19, 20]. The non-toxicity and safety of long-term daily administration of Glakay$^R$ have already been well established and appear to be clinically beneficial in treating elderly patients [19, 21].

In the present study, we investigated the combined effects of VK2 in elderly AML patients treated with VEN and AZA. We observed that the incidence of complete remission (CR) was extremely high in patients administered VK2 during treatment with VEN and AZAs. Furthermore, preclinical results indicated that VK2 suppresses MCL-1 through NOXA induction. NOXA is a pro-apoptotic BH3-only protein that interacts with and suppresses MCL-1. The combined results of the cell-based and clinical studies suggest that VK2 exhibits sufficient activity against patients with AML and warrants consideration for combined use with VEN and hypomethylating agents.

## Materials and methods

### Protocol and patients

Between October 1, 2021, and April 30, 2022, 19 patients who had already diagnosed with AML and treated with azacitidine, venetoclax, and vitamin K2 (diagnosed after April 1, 2021) or newly diagnosed with AML were enrolled in this study. All patients received vitamin K2 (45 mg/day) as osteoporosis therapy during treatment. The patients received azacitidine 75 mg/m$^2$ on Day 1–7, and venetoclax 400 mg on Day 1–28. Patients were monitored for response and toxicity according to their intent-to-treat. The study was conducted in accordance with the International Conference on Harmonization, Good Clinical Practice Guidelines, and the Declaration of Helsinki. The Institutional Ethics Committee of Shinyurigaoka General Hospital approved the trial protocol (#STR-3-9-29). Written informed consent was obtained from all patients. For consolidation, some patients were sequentially treated with Gemtuzumab and Ozogamicin (cases 2 and 12) or gilteritinib (case 13).

### Reagents

Venetoclax (VEN) was obtained from MedChemexpress Co. (Monmouth Junction, NJ, USA) and dissolved in dimethyl sulfoxide (DMSO; Nacalai Tesque, Kyoto, Japan) to prepare 10 mM stock solutions. Vitamin K2 (VK2) (menaquinone-4; KaytwoN Intravenous Injection$^R$ containing 10 mg MK-4 in each vial) was obtained from Eisai Co. (Tokyo, Japan). KaytwoN intravenous Injection$^R$ was diluted with RPMI-1640 culture medium to obtain a 1 mM solution. As VK2 is photosensitive and easily degraded by light, it was prepared for each experiment without preparing a stock solution. All untreated controls were supplemented with DMSO to match the volume of VEN treatments. N-acetyl-l-cysteine (NAC), melatonin, 5-Azacytidine (5-Aza), and carbonyl cyanide m-chlorophenylhydrazone (CCCP) were obtained from Nacalai Tesque. Trolox was obtained from the Tokyo Chemical Industry (Tokyo, Japan).

### Cells and cell culture

Human acute promyelocytic leukemia-derived HL-60, human acute monocytic leukemia-derived THP-1, and human histiocytic lymphoma-derived U-937 cells were obtained from American Type Culture Collection (ATCC, Manassas, VA, USA). Human AML-derived MOLM-14 and SKM-1 cells were obtained from JCRB Cell Bank (Osaka, Japan). All cell lines were cultured with RPMI1640 (Sigma-Aldrich, St. Louis, MO, USA) supplemented with 10% fetal bovine serum (Gibco; Thermo Fisher Scientific, Inc., Waltham, MA, USA) and 1% penicillin/streptomycin solution (FUJIFILM Wako Pure Chemical, Tokyo, Japan) in a humified 5% $CO_2$ incubator at 37˚C. All cell line experiments were conducted within 10 passages after thawing. Mycoplasma contamination was tested routinely using the e-Myco Mycoplasma PCR Detection kit ver.2.0 (iNtRON Biotechnology, Inc., Korea).

### Establishment of knockout cells

NOXA-KO SKM-1 and HL-60 cells were established as previously described [22]. SKM-1 and HL-60 cells were transfected with pSpCas9 (BB)-2A-Puro (PX459) V2.0 plasmid (Addgene, #62988) with guide sequence for negative control (sgNega: 5′−CACCGGTAGCGAACGTG TCCGGCGT−3′) or *NOXA* (sgNOXA: 5′−CACCGCGGCACCGGCGGAGATGCCT−3′) by electroporation with Super Electroporator NEPA 21 with a 2 mm gap cuvette (cat. no. EC 002) (NEPA GENE Co., Ltd., Chiba, Japan). The following conditions were used: Poring pulse (voltage, 275 V; pulse interval, 1.5 ms; pulse width, 50 ms; pulse number, 2; attenuation rate, 40%); and transfer pulse (voltage, 20 V; pulse interval, 50 ms; pulse width, 50 ms; pulse

number, 5; and attenuation rate, 40%). From the next day, cells were treated with 0.5 μg/ml puromycin for 3 days and surviving cells were used in the following experiments.

## Knockdown experiments

For the gene silencing of NOXA in SKM-1 cells and HL-60 cells, NOXA siRNA and control siRNA were synthesized as follows (FASMAC): siNOXA: sense `GUAAUUAUUGACACAUUU CdTdT` and antisense `GAAAUGUGUCAAUAAUUACdTdT` [23]; siControl: sense `GUUAAAG GUUUGACUCGCGdTdT` and antisense `CGCGAGUCAAACCUUUAACdTdT` [24]. siRNA was diluted to 1500 nM in 100 μL Opti-MEM (Thermo) containing $1 \times 10^6$ cells. Transfection was performed by electroporation with a Super Electroporator NEPA 21 with a 2 mm gap cuvette. The experimental conditions were the same as those described above. Twenty-four hours after the transfection, the cells were treated with VK2, VEN, or VK2 plus VEN for 48 h and used for cell viability assay or immunoblotting as described below.

## Cell viability assay

Cell viability was assessed using the CellTiter Blue Cell Viability Assay Kit (Promega, Madison, WI, USA) according to the manufacturer's instructions with HL-60 (wild-type and NOX-A-KO), SKM-1 (wild-type and NOXA-KO), THP-1, U-937, and MOLM-14 cell lines. Briefly, $2 \times 10^4$ cells/ well were seeded into a 96-well plate with venetoclax ±VK2 and cultured for 48 or 72 h. CellTiter Blue reagent was added to each well and fluorescence was measured (excitation, 560 nm; emission, 590 nm) using a SpectraMax iD3 fluorometer (Molecular Devices, LLC, San Jose, CA, USA). The mean fluorescence relative to that of untreated cells was expressed as a percentage of cellular proliferation.

## Apoptosis assay

To assess apoptosis, HL-60 and SKM-1 cells were treated with VK2 with or without VEN for 48 h and resuspended in Annexin V-binding buffer at $1 \times 10^6$ cells/ml and stained using the Annexin V-FITC Apoptosis Detection Kit (Nacalai Tesque, Inc.) according to the manufacturer's instructions. Flow cytometry was performed using the Attune Acoustic Focusing Cytometer. Data analysis was performed using the Attune Cytometric Software v2.1 (Life Technologies, Carlsbad, CA, USA).

## May-Grünwald-giemsa staining

HL-60 and SKM-1 cells were treated with either VK2, VEN, or VK2 plus VEN for 48 h and then cell spreads were prepared on glass slides using a Cytospin 4 centrifuge (Thermo Fisher Scientific, Inc.) at $1,000 \times g$ for 5 min at room temperature (RT). May-Grünwald-Giemsa staining was performed using the May-Grünwald staining solution (without dilution; cat. no. 15053; Muto Pure Chemicals, Tokyo, Japan) for 3 min at RT followed by Giemsa staining (1 drop/1 ml $H_2O$; cat. no. 15003; Muto Pure Chemicals, Tokyo, Japan) for 15 min at RT. The glass slides were examined under a digital light microscope (BZ-800; Keyence Corporation, Osaka, Japan). Representative images were selected for further analysis.

## Immunoblotting

HL-60 (wild-type or NOXA-KO) and SKM-1 (wild-type or NOXA-KO) cells were treated with either VK2, VEN, or VK2 plus VEN for 24 h or 48 h. Then, the cells were lysed using RIPA buffer (Nacalai Tesque) containing a protease and phosphatase inhibitor cocktail (Nacalai Tesque). Equal amounts of protein were separated by sodium dodecyl sulfate-

polyacrylamide gel electrophoresis (SDS-PAGE), and transferred onto Immobilon-P membranes (Millipore Corp., Bedford, MA, USA). These membranes were probed with primary antibodies, such as anti-NOXA Ab (sc-56169, 1:1,000), anti-GAPDH Ab (sc-32233, 1:1,000), anti-BAX Ab (sc-493, 1:1,000), (Santa Cruz Biotechnology, Santa Cruz, CA, USA), anti-BCL-2 Ab (#551097, 1:1,000, BD Biosciences s, Bedford, MA), anti-BAK Ab (06–536, 1:1,000) (Merk Millipore, Burlington, MA, USA), anti-PARP Ab (#9542, 1:1000), anti-cleaved Caspase-3 Ab (#9661, 1:1000), anti-Mcl-1 Ab (#5453, 1:1000), anti-BCL-XL Ab (#2764, 1:1000) (Cell Signaling Technology, Danvers, MA, USA) for 16 h at 4°C. The immunoreactive proteins were detected using horseradish peroxidase-conjugated secondary antibodies (cat. no. 115-035-003; 1:2,500; or anti-rabbit; cat. no. 711-035-152; 1:2,500; both from Jackson ImmunoResearch) for 1 h at room temperature. Immobilon Western Chemiluminescent HRP Substrate (cat. no. WBKLS0500; Merck Millipore) was added and protein expression was detected using a WSE-6300H/C Luminograph III (ATTO). **Uncropped images of western blotting are shown in S3 Fig.**

### Realtime PCR

Total RNA was extracted using the NucleoSpin RNA Plus (Takara Bio Inc. Otsu, Japan). cDNA was synthesized using the PrimeScript RT Master Mix (Takara Bio, Inc.). To assess the gene expression levels, qPCR was performed using TB Green Ex Taq II (Tli RNase H Plus) (Takara Bio, Inc.). The target mRNA expression level was calculated, relative to GAPDH expression levels in the same sample, based on the $\Delta(\Delta Ct)$ method. Following primers were used; NOXA-F GGAGATGCCTGGGAAGAAG, NOXA-R CCTGAGTTGAGTAGCACACTCG; MCL-1-F AAGCCAATGGGCAGGTCT, MCL1-R TGTCCAGTTTCCGAAGCAT; BCL-2-F TCAGCATGGCTCAAAGTGCAG, BCL-2-R GAAACAGATGTCCCTACCAACCAGA; BAX-F CATGGGCTGGACATTGGACT, BAX-R GAGAGGAGGCCGTCCCAA; BAK-F CCATTCCTGGAAACTGGGCT, BAK-R GACGGGATCAGCCTGCC; BCL-XL-F AAAAGATCTTCCGGGGGCTG, BCX-XL-R TCTGAAGGGAGAGAAAGAGATTCA; GAPDH-F GCACCGTCAAGGCTGAGAAC, GAPDH-R TGGTGAAGACGCCAGTGGA. Quantitative real-time PCR was performed using a CFX Opus 96 Real-Time PCR System (Bio-Rad). Data analysis was performed using the CFX Maestro software Ver 2.3 (Bio-rad).

### Measurement of ROS

Whole-cell ROS levels were determined using dihydroethidium (DHE) (Calbiochem). HL-60 or SKM-1 cells were treated with VEN ± VK2 for 48 h and then 5 μM DHE was added into the culture medium and incubated for 15 min in $CO_2$ incubator. The cells were then washed with PBS and observed under a digital light microscope (BZX-800; Keyence Corporation). Mitochondrial ROS levels were determined using MitoSox-Red staining. HL-60 cells and SKM-1 cells were treated with VEN ± VK2 for 24 or 48 h and then incubated with 2.5 μM MitoSox Red in Hanks Balanced Salt Solution (HBSS; FUJIFILM Wako Pure Chemical Corp.) for 30 min in $CO_2$ incubator. After washing with PBS, the cells were resuspended in PBS and assessed using a flow cytometer (Attune Acoustic Focusing Cytometer; Life Technologies). Healthy cells gated with forward scatter (FSC) and side scatter (SSC) of flow cytometry were used for the analysis. Data analysis was performed using the Attune Cytometric Software v2.1 (Life Technologies).

### Measurement of mitochondrial membrane potential

Mitochondrial membrane potential was assessed using tetramethylrhodamine ethyl ester (TMRE) (Thermo Fisher Scientific). HL-60 and SKM-1 cells were treated with a vehicle,

VEN ± VK2 ± NAC, or 10 μM CCCP for 24 h. After treatment, the TMRE probe was added to each cell line (final concentration, 250 nM). The cells were incubated in a $CO_2$ incubator for 30 min, harvested, and washed twice with PBS. The TMRE fluorescence was measured using an Attune flow cytometer (Thermo Fisher Scientific). Similar to the mitochondrial ROS measurement, healthy cells were gated for analysis.

### Statistical analysis

The synergistic effect of VK2 and VEN on the inhibition of cellular proliferation was statistically analyzed using Combenefit software ver. 2.021 [25]. All quantitative data are expressed as the mean ± standard deviation. One-way ANOVA followed by post-hoc tests were used to compare multiple groups, and pairwise comparisons were performed using Tukey's honest significant difference test (if all groups were to be compared) or the Games-Howell test whenever all groups were compared with a single reference group. $p < 0.05$ was considered to indicate a statistically significant. All raw data used for graphs are available in S1–S5 Files.

## Results

### High CR/CRi rates by AZA plus VEN in the AML patients who received daily VK2

Nineteen AML patients were enrolled in this study. All patients received oral VK2 (45 mg/day) during the course of AZA plus VEN chemotherapy. Patient characteristics are shown in Table 1. The median age was 75.9 years (range 66–84). Most patients had high-risk features: 13 patients (68.4%) had ELN (European Leukemia Network) adverse risk, 47% ECOG PS (Eastern Cooperative Oncology Group Performance Status) 2 >, and 2 (10.5%) had TP53 deletion. The patients were evaluated for response and toxicity according to their intention-to-treat status. The complete remission (CR) with incomplete count recovery (CRi) rate was 94.7% (18/19), with a CR rate of 79% (Table 2). The eight-week mortality was 0%. All patients

**Table 1. Characteristics of 19 AML patients enrolled in the study.**

| Characteristics | N = 19 (%) |
|---|---|
| **Age, median (range)** | 75.9 (66–84) |
| **ECOG PS** | |
| 0 | 2 (10.5) |
| 1 | 8 (42.1) |
| 2 | 7 (36.8) |
| 3 | 2 (10.5) |
| **Mutations** | |
| NPM1 | 3 (15.7) |
| FLT-3TKD/ITD | 1 (5.2) |
| TP53 | 2 (10.5) |
| KIT | 1 (5.2) |
| **ELN Risk** | |
| Favorable | 1 (5.2) |
| Intermediate | 5 (26.3) |
| Adverse | 13 (68.4) |
| **Complex Karyotype** | 9 (47.7) |

Abbreviations: N, number; ECOG PS, Eastern Cooperative Oncology Group Performance Status; ELN, European Leukemia Network.

**Table 2. Responses of 19 AML patients enrolled in the study.**

| Results | N = 19 (%) |
|---|---|
| **ORR** | 19 (100) |
| CR/CRi | 18 (95) |
| CR | 15 (79) |
| CRi | 2 (10) |
| MLFS | 1 (5) |
| **No Response** | 0 (0) |
| **Time to first response (months)** | 0.8 |
| **Median time to ANC>0.5 (days)** | 35 |
| **Median time to platelet>50 (days)** | 28 |
| **4-week mortality** | 0 (0) |
| **8-week mortality** | 0 (0) |

Abbreviations: N, number; ORR, overall response rate; CR, complete remission; CRi, complete remission with incomplete count recovery; MLFS, morphological leukemia-free state; ANC, absolute neutrophil count.

achieved a response after cycle 1. The median times to absolute neutrophil count (ANC) recovery ($> 0.5$) and platelet recovery (50) among the CR/CRi patients were 35 days and 28 days, respectively. Complete cytogenetic CR was achieved in 15 of 19 (79%) evaluable and minimal residual disease (MRD) negativity in 2 of 15 (13%) evaluable CR patients. After a median follow-up, one patient (case 16) with TP53 deletion had CNS relapse and died 15.4 months after the beginning of treatment, and five patients died 15.6, 16.6, 17.8, 19.2, and 26.4 months after the beginning of treatment after bone marrow relapse respectively (**Fig 1**). Although overall survival in this study was similar to that in previous reports, an extremely high incidence of complete remission was observed in AML patients treated with VEN, AZA, and VK2.

## Enhanced apoptosis induction in AML cells by combined treatment with VK2 plus VEN

The extremely high CR/CRi rate in patients with AML who received AZA plus VEN therapy with daily oral VK2 prompted us to investigate the underlying molecular mechanisms. Our previous *in vitro* study showed that VK2 induces apoptosis in AML cells with low BCL-2 expression and induces differentiation in AML cells with higher BCL-2 expression [14]. Because all patients have simultaneously received a specific BCL-2 inhibitor, VEN, with VK2 (45 mg/day, po) during the entire cycle of chemotherapy, we hypothesized that the drug combination of VEN plus VK2 rather than AZA plus VK2 might be attributed to the higher response rate after cycle 1. Therefore, we focused on the combined effects of VEN and VK2 in AML cell lines. As previously reported, VK2 inhibited the growth of AML cell lines (**Fig 2**) [13, 14, 20]. In all five AML cell lines tested, simultaneous treatment with VK2 and VEN resulted in synergistically enhanced cell growth inhibition compared to cells treated with VEN or VK2 alone (**Fig 2**). The slightly weaker synergistic effect in MOLM-14 cells compared to that in other cell lines appeared to be due to the high sensitivity to VK2 monotherapy (**Fig 2E**). In addition, the combination of AZA and VK2 did not show synergistic effects in contrast to the combination of VEN and VK2 (S1 Fig). These data suggested that VK2 invariably sensitizes AML cells to VEN.

After treatment with VEN and VK2, HL-60 and SKM-1 cells showed nuclear fragments, apoptotic bodies, and chromatin condensation, which are typical morphological features of cells undergoing apoptosis (**Fig 3A and 3B**). We also observed an increase in annexin-V-

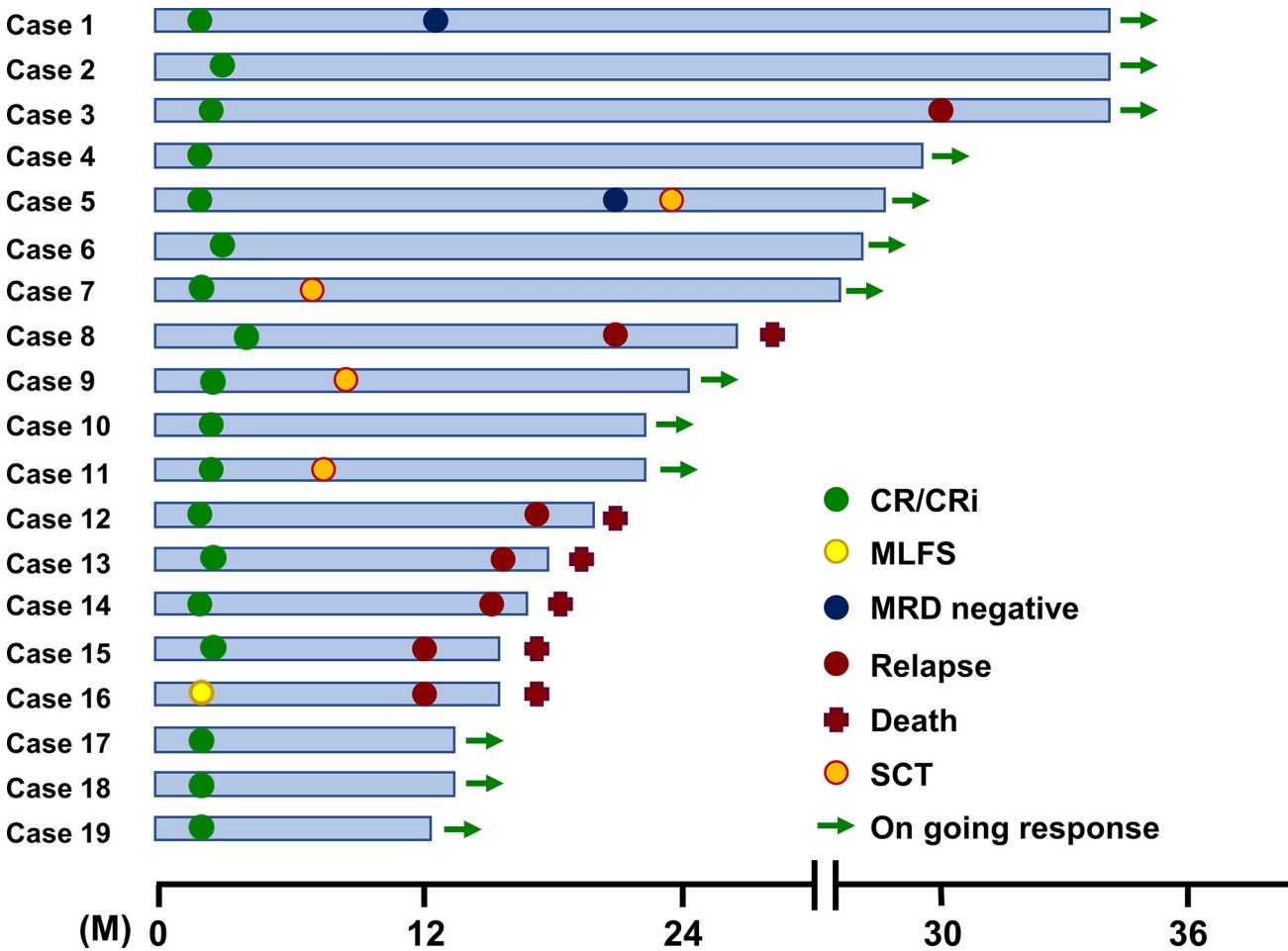

**Fig 1. Therapeutic outcome of 19 AML patients receiving VEN+AZA+VK2.** CR, complete response; CRi, CR with incomplete count recovery; MLFS, morphologic leukemia-free state; MRD, minimal residual disease; M, months after the initiation of AZA+VEN treatment.

positive/PI-positive cell numbers by flow cytometry (**Fig 3C and 3D**) and increased expression of cleaved caspase-3 as well as cleaved PARP by immunoblotting after treating the cells with VEN plus VK2 as compared with those treated with VEN or VK2 alone (**Fig 3E and 3F**). These data show enhanced apoptosis induction by the two-drug combination. We also assessed the expression of apoptosis-related proteins by immunoblotting after treatment with VEN and/or VK2 for 24 h and 48 h (**Fig 3E and 3F**). Notably, the expression of the proapoptotic BH3-only protein, NOXA, was clearly increased after treatment with VK2 and VK2 plus VEN, whereas the expression of other BCL-2 family proteins, including MCL-1, decreased after 48 h of exposure to VEN plus VK2. VEN monotreatment did not increase NOXA levels in either cell line. These data indicated that VK2, but not VEN, induces NOXA expression.

### Reactive oxygen species (ROS) production in response to VK2 leads to the pronounced cytotoxicity of VEN with up-regulation of NOXA and down-regulation of MCL-1

Previous studies have shown that ROS is responsible for VK2-induced cytotoxicity. [16, 26–29]. DHE staining showed that VK2, but not VEN, enhanced ROS production in both the cell lines (**Fig 4A and 4B**). Additionally, MitoSOX red staining confirmed that these ROS were

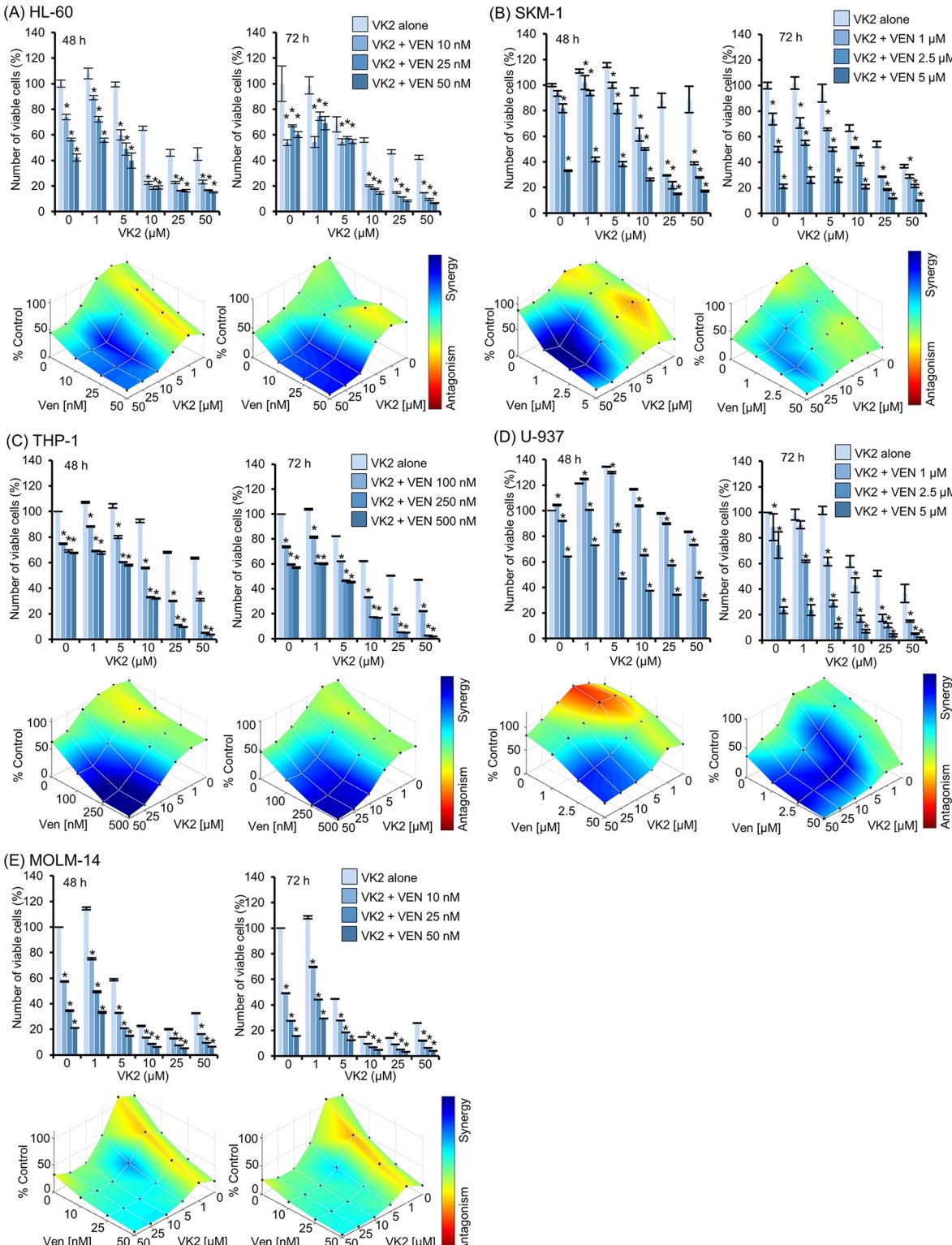

**Fig 2. Enhanced cell growth inhibition by combination treatment of VK2 and VEN in AML cell lines.** (A-E) AML cell lines (HL-60, SKM-1, THP-1, U-937, and MOLM-14) were treated with VK2 in the presence or absence of VEN at indicated concentrations for 48 h and 72 h. Upper: The viable cell number was assessed by CellTiter Blue assay. Data are presented as the mean ± SD. *p<0.05 vs. VEN 0 nM. Lower: The synergistic effect of VK2 and VEN combined treatment on AML cell proliferative inhibition was statistically analyzed using Combenefit software. Mapping of the synergy levels on the experimental combination dose-response surface. A higher score shown in denser blue indicates a stronger synergistic effect. n = 3.

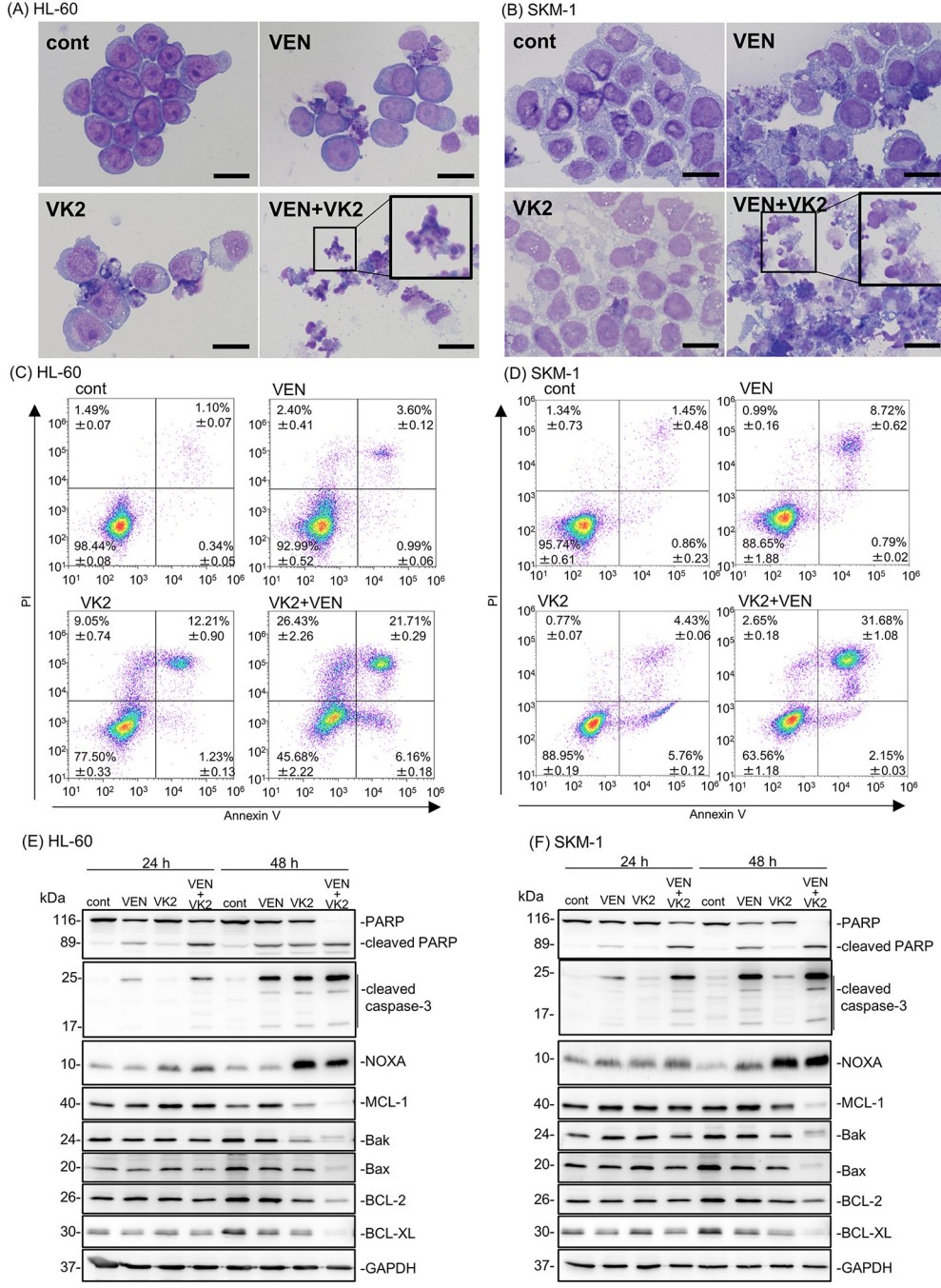

**Fig 3. Pronounced apoptosis induction by simultaneous treatment with VK2 and VEN along with up-regulation of NOXA and repression of MCL-1 in HL-60 and SKM-1 cells.** (A, B) HL-60 and SKM-1 cells were treated with either VK2 (10 μM for HL-60, 25 μM for SKM-1), VEN (25 nM for HL-60, 2.5 μM for SKM-1), or VK2 plus VEN for 48 h, and stained with May-Grunwald-Giemsa. Scale bar = 20 μm. (C, D) Cells were treated with either VK2 (10 μM for HL-60, 25 μM for SKM-1), VEN (25 nM for HL-60, 2.5 μM for SKM-1), or VK2 plus VEN for 48 h. Flow cytometry was performed with Annexin V and PI double staining. The number of each area indicates the percentage of cells. n = 3 (E, F) Cellular proteins were lysed, separated by SDS-PAGE, and immunoblotting was performed using indicated antibodies. Immunoblotting with anti-GAPDH mAb was performed as an internal loading control.

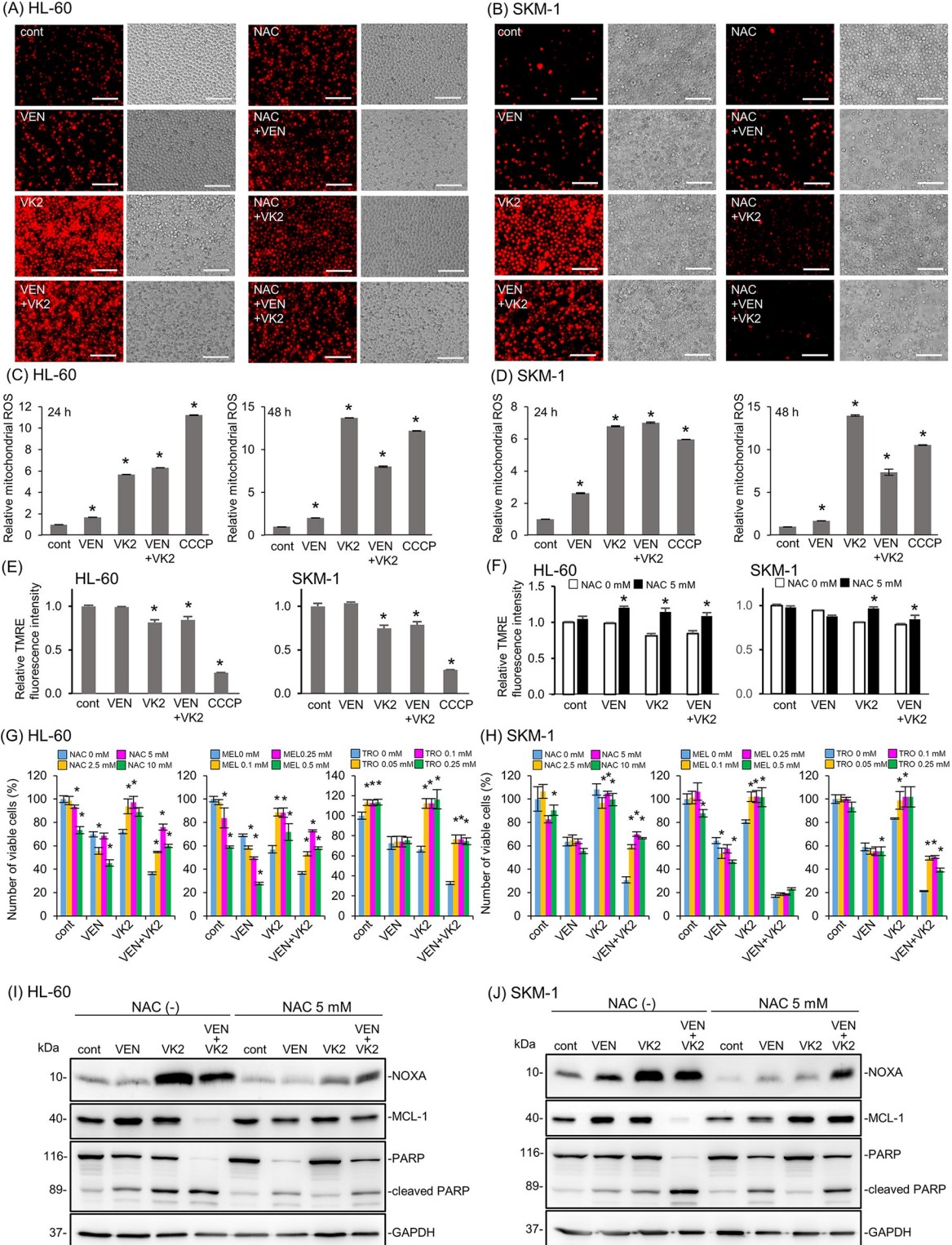

**Fig 4. ROS production in response to VK2 and/or VEN in HL-60 and SKM-1 cells.** Cells were treated with VK2 (10 μM for HL-60, 25 μM for SKM-1) or VEN (25 nM for HL-60, 2.5 μM for SKM-1) for 24 and 48 h. (A, B) ROS production in whole cells was assessed by staining with dihydroethidium (DHE) and detected by fluorescence microscopy in HL-60 and SKM-1 cells 48 h after treatment. Scale bar = 100 μm. (C, D) Mitochondrial ROS levels were determined using flow cytometry after staining with MitoSox-Red. The cells treated with a mitochondrial uncoupler, CCCP (10 μM) were used as a positive control for mitochondrial ROS production. Data are presented as

the mean ± SD. n = 3, *p<0.05 vs. cont. (E) The mitochondrial membrane potential was assessed by flow cytometry after TMRE staining. CCCP was used as the positive control. Data are presented as the mean ± SD. n = 3, *p<0.05 vs. cont. (F) The effect of ROS scavengers on the mitochondrial membrane potential was assessed by flow cytometry. Data are presented as the mean ± SD. n = 3, *p<0.05 vs. 0 mM NAC. (G, H) Cells were treated with VK2 (10 μM for HL-60, 25 μM for SKM-1) and/or VEN (25 nM for HL-60, 2.5 μM for SKM-1) in the presence of ROS scavengers, namely NAC, melatonin (MEL), and Trolox (TRO) at the indicated concentrations for 48 h. The number of viable cells was assessed using the CellTiter Blue assay. Data are presented as the mean ± SD. *p<0.05 vs. 0 mM NAC, MEL, or TRO. (I, J) Immunoblotting with anti-NOXA, anti-MCL-1, and anti-PARP mAbs. Immunoblotting with anti-GAPDH mAb was performed as an internal loading control.

derived from mitochondria (**Fig 4C and 4D**). The combination of VK2 and VEN did not further increase mitochondrial ROS production compared with VK2 alone (**Fig 4C and 4D**). The VK2-induced reduction in the mitochondrial membrane potential was moderate and was restored by NAC treatment (**Fig 4E and 4F**). This suggests that mitochondrial ROS were not produced due to mitochondrial dysfunction, but rather that VK2-induced ROS damage the mitochondria. Similar results have also been observed in bladder cancer cells [30].

Next, we treated HL-60 and SKM-1 cells with VK2 and VEN in the presence of ROS scavengers, namely N-acetyl-L-cysteine (NAC), melatonin, and Trolox. Except for SKM-1 cells treated with melatonin, all ROS scavengers canceled or repressed the pronounced cytotoxicity induced by the VK2 plus VEN combination to the extent of VEN mono-treatment in both cell lines (**Fig 4G and 4H**). Notably, these ROS scavengers canceled the enhanced cytotoxicity induced by VK2 but had no or moderate effects on the cytotoxicity of VEN. Thus, VK2-induced ROS are essential for the pronounced induction of apoptosis in response to the combination of VK2 and VEN.

As shown in **Fig 3E and 3F**, the anti-apoptotic protein MCL-1 was suppressed after 48 h of treatment with VK2 and VEN. It is well known that NOXA determines the localization and stability of MCL-1 on the mitochondrial outer membrane [31]. NOXA physically interacts with MCL-1, leading to MCL-1 degradation via polyubiquitination [32]. Thus, NOXA is a determinant for MCL-1 expression [33, 34]. Additionally, the expression of MCL-1 is one of a determinant of VEN sensitivity in AML cells [35]. Therefore, we investigated the effect of ROS on NOXA and MCL-1 expression in VK2-treated AML cells. The presence of NAC attenuated NOXA induction along with the cancelation of MCL-1 suppression and PARP cleavage in VK2- and VK2 + VEN-treated cells (**Fig 4I and 4J**). These data suggested that mitochondrial ROS production in response to VK2 treatment is critical upstream of the NOXA-MCL-1 axis to exert pronounced cytotoxicity.

## NOXA is crucial for the synergistic apoptosis induction by concomitant VEN and VK2 treatment in AML cells

Our results showed an increased expression of NOXA in response to VK2 treatment in AML cells (**Figs 3E, 3F, 4I and 4J**). Gene expression profiles of BCL-2 family proteins showed that VK2 treatment induced more prominent transcriptional activation of *NOXA* than other BCL-2 family genes in HL-60 and SKM-1 cells (**Fig 5A and 5B**). Concomitant exposure to VK2 and VEN did not increase *NOXA* expression. The VK2-induced transcriptional activation of *NOXA* was repressed in the presence of NAC in HL-60 and SKM-1 cells, similar to protein expression, indicating that the induction of NOXA was regulated at the transcriptional level (**Fig 5C and 5D**). Finally, we confirmed the importance of NOXA induction in synergistic cell death caused by the co-administration of VK2 and VEN. Knockdown (KD) and knockout (KO) of *NOXA* in HL-60 and SKM-1 cells resulted in the abrogation and attenuation of synergistic cell death by VK2 and VEN, as well as the repression of PARP cleavage (**Fig 6A–6D, S2 Fig**). After treatment with VK2 and VEN, control SKM-1 and HL-60 cells exhibited MCL-1

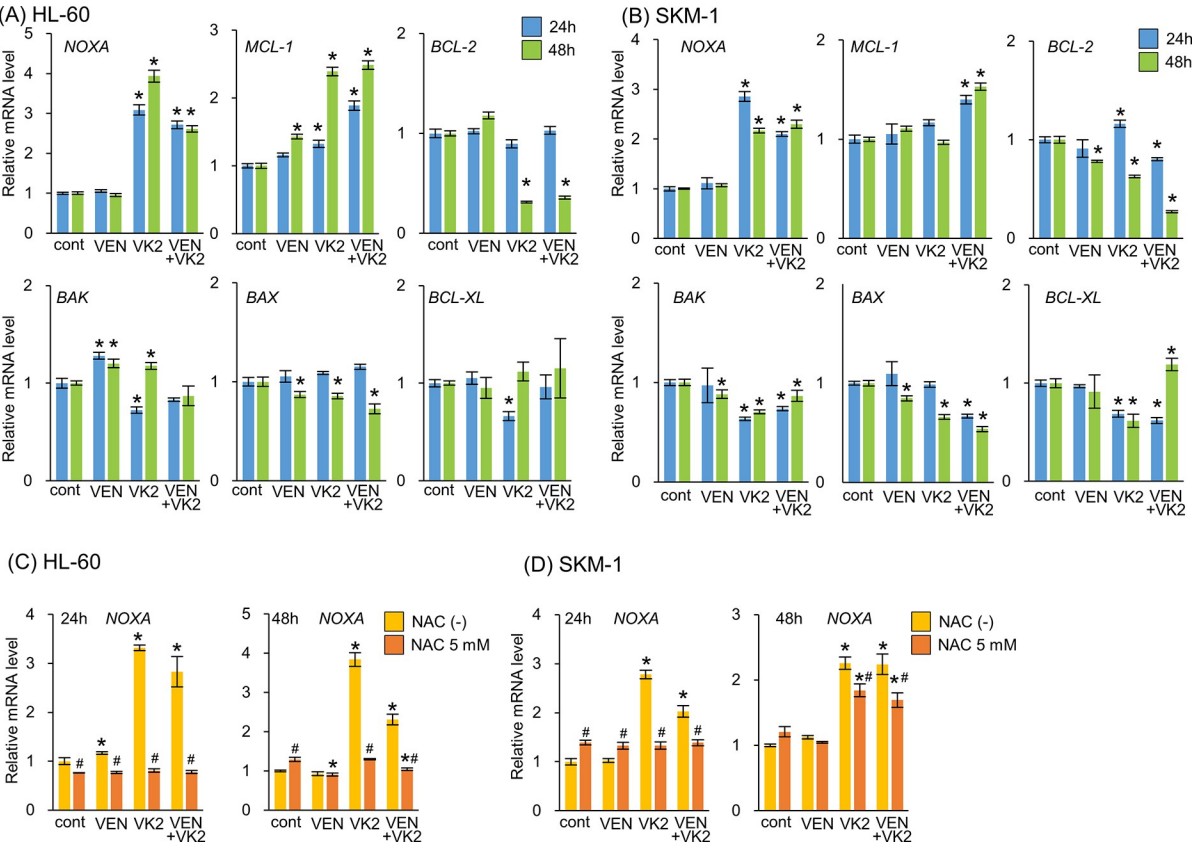

**Fig 5. Gene expressions of BCL-2 family members after treatment with VK2 and/or VEN in HL-60 and SKM-1 cells.** (A, B) HL-60 and SKM-1 cells were treated with VK2 (10 µM for HL-60, 25 µM for SKM-1) and/or VEN (25 nM for HL-60, 2.5 µM for SKM-1) for 24 h and 48 h. Gene expressions of the BCL-2 family members were assessed by real-time PCR. Data are presented as the mean ± SD. *p<0.05 vs. cont. n = 3. (C, D) Effect of NAC treatment on *NOXA* mRNA expression was assessed by real-time PCR with HL-60 and SKM-1 cells. Data are presented as the mean ± SD., *p<0.05 vs. cont, #p<0.05 v.s. NAC (-), n = 3.

repression along with NOXA upregulation, whereas KD and KO of *NOXA* in both cell lines attenuated MCL-1 repression in response to VK2 and VEN (**Fig 6C and 6D, S2 Fig**). Taken together, we conclude that NOXA upregulation in response to VK2 is crucial for synergistic apoptosis induction by VEN and VK2.

## Discussion

The extremely high CR/CRi rate in 19 elderly AML patients who received AZA plus VEN therapy with daily oral Glakay[R] (VK2) prompted us to investigate the underlying molecular mechanisms. Simultaneous exposure to VEN and VK2 synergistically induced apoptosis in all AML cell lines tested (Figs 2 and 3). VK2, but not VEN, induced ROS-mediated transcriptional activation of NOXA (Figs 4 and 5). Repression of NOXA induction by ROS scavengers, as well as KD and KO of NOXA, resulted in almost complete cancellation of the synergistic cell death by the VEN + VK2 combination (Figs 5 and 6). Thus, as shown in Fig 6E, ROS-mediated NOXA induction appears to be crucial for enhancing apoptosis in AML cells. Transcriptional upregulation of NOXA by ROS has also been reported in CLL cells [37]. Notably, along with NOXA upregulation, the repression of the anti-apoptotic MCL-1 protein was always accompanied by a series of experiments when AML cells were treated with VEN plus VK2 (Figs 3, 4, and 6). Since VEN is a BCL-2-selective inhibitor with Ki <0.01 nM and no activity against MCL-1 in

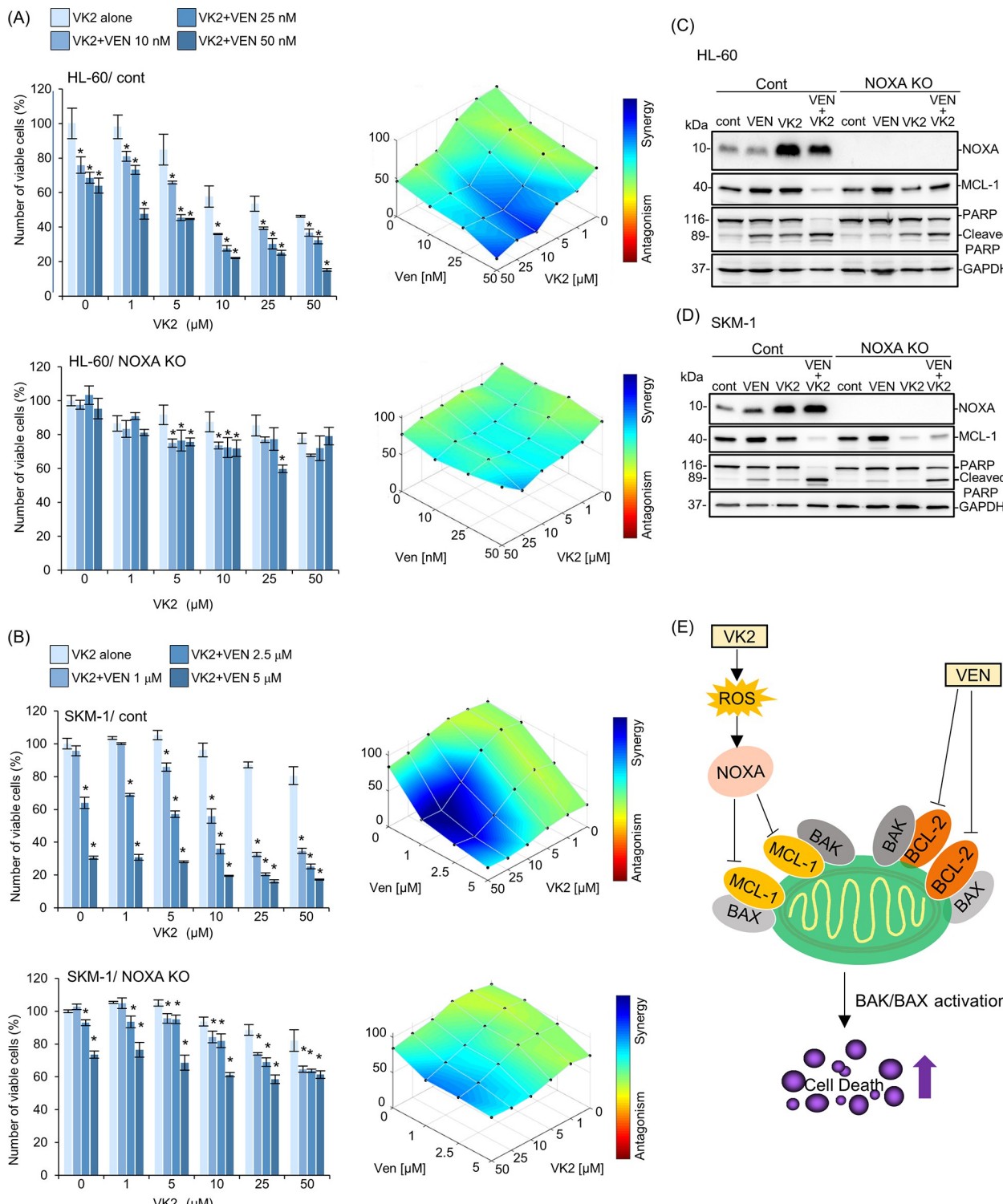

**Fig 6. Effect of NOXA-knockout in HL-60 and SKM-1 cells on the cytotoxicity and MCL-1 repression by VK2 and VEN combination treatment.** (A, B) Control and NOXA knockout (KO) HL-60 and SKM-1 cells were treated with VK2 with/without VEN at indicated concentrations, and the viable cell numbers were assessed by CellTiter Blue assay. Data are presented as the mean ± SD. *p<0.05 vs. VEN 0 nM. The synergistic effect of VK2 and VEN combined treatment on each cell proliferative inhibition was statistically analyzed using Combenefit software. (C, D) After treatment with VK2 with/without VEN for 48 h, cellular proteins were separated by SDS-PAGE and immunoblotted with anti-NOXA, anti-MCL-1, and anti-PARP Abs. Immunoblotting with anti-GAPDH mAb was performed as an internal control. (E) Proposal scheme of the molecular mechanism of VK2 for sensitization to VEN in AML cells. VEN-treatment specifically inhibits BCL-2 but not MCL-1 [36].

VK2-treatment induces mitochondrial ROS production leading to up-regulation of NOXA, which results in inhibition and/or repression of MCL-1 to relieve BAK inhibition. Thus, concomitant treatment with VEN plus VK2 results in simultaneous inhibition of BCL-2 and MCL-1. This appears to enhance BAK- and/or BAK/BAX-mediated apoptosis induction in AML cells.

cell-free assays [36], VK2-mediated MCL-1 repression appears to compensate for the full activation of BAK and BAX, which are responsible for forming oligomeric pores on the mitochondrial outer members to release cytochrome c for apoptosis induction [38, 39]. Previous studies showed that MCL-1 expression is a determinant of VEN sensitivity in AML patients [35]. Additionally, the ratio of MCL-1 to BCL-2 expression determined the response to VEN in myeloma cell lines [40]. Upregulation of MCL-1 was observed in VEN-resistant AML and MDS, and the combination of VEN and MCL-1 inhibitor was synergistic in all MDS subtypes without significant injury to normal hematopoiesis in humanized MISTRG6 mice [6]. Additionally, the dual inhibition of BCL-2 and MCL-1 improves the therapeutic efficiency of BH3-mimetics in AML cell lines [41]. NOXA binds to MCL-1 and disrupts the interaction between BAK and MCL-1, thus relieving BAK inhibition [1, 34, 42, 43]. Alternatively, NOXA physically interacts with MCL-1, which leads to its MCL-1 degradation in mitochondria via polyubiquitination by the E3 ligase Mule [32]. This indicates that NOXA upregulation by VK2, as shown in this study, is sufficient for MCL-1 inhibition, regardless of the decrease in MCL-1. Therefore, the ROS-NOXA-MCL-1 axis in response to VK2 appears to be attributed to VEN sensitization in AML cells, which appears to be the underlying molecular mechanism of the high response rate in our clinical outcomes (**Table 2, Fig 1**). Recent research on transcriptional and phenotypic heterogeneity within or among patients has shown that resistance to venetoclax in AML is not solely dependent on MCL1 expression [44]. Further studies is needed to identify the type of AML sensitized by VK2 in combination with venetoclax.

Although we have shown the underlying mechanism of enhanced apoptosis induction in terms of the dual inhibition of BCL-2 by VEN and MCL-1 via NOXA upregulation by VK2, there is another possibility. We previously showed that the VK2-2,3 epoxide, an intracellular metabolite of VK2 induced by gamma-glutamyl carboxylase (GGCX), covalently binds to Bak, directly leading to Bak-mediated apoptosis in leukemia cells [28]. Additionally, hypomethylating agents, including AZA, have been shown to reduce the levels of MLC-1 in primary AML cells and induce a DNA damage response, leading to the upregulation of BH3-only proteins such as NOXA [45]. Thus, in clinical applications, the therapeutic effects of the combination of VK2 and AZA should also be discussed.

Consistent with our results, many reports show that ROS production is involved in VK2-induced apoptotic and non-apoptotic cell death in various kinds of cancer cells [26, 27, 29, 30]. Although it is not clear how VK2-derived ROS are produced, it is well established that VK2 is metabolized in the vitamin K cycle, where it is located in the endoplasmic reticulum (ER) and undergoes electron reduction to a reduced form called vitamin K hydroquinone, catalyzed by vitamin K epoxide reductase (VKOR) or ferroptosis suppressor protein 1 (FSP1) [46, 47]. Subsequently, gamma-glutamyl carboxylase oxidizes vitamin K hydroquinone to vitamin K epoxide and this reaction is coupled with carboxylation of Glu to Gla. The VK epoxide is then converted to vitamin K by VKOR [48]. However, according to our data, ROS levels increased with VK2 treatment in the mitochondria instead of the ER. Although the ER is distributed inside the cell to form a network with organelles, including the mitochondria, the precise mechanism of ROS production in response to VK2 remains unclear.

Regarding the pharmacokinetics of VK2, the Cmax of 15 mg of orally administered menaquinone-4 (Glakay$^R$) has been reported to be approximately 1 μM [49], which is one-tenth that of our *in vitro* study shown in Fig 2. However, VK2 is a fat-soluble vitamin that has been

shown to be more than 10 times higher in the bone marrow than in the blood in a murine model [50, 51]. Thus, the concentration of VK2 that synergizes with VEN for apoptosis induction observed in our study appears to be physiological in clinical settings.

Although the clinical outcomes presented here are based on a small number of patients from a single institute, the high CR rate and tolerability in unfavorably patients with AML are noteworthy. Since VK2 (Glakay$^R$) itself is nontoxic and has been administered daily for a long time to patients with osteoporosis in Japan, the use of VK2 as a chemosensitizer for VEN via NOXA-mediated MCL-1 suppression appears to be a promising strategy. We emphasize that a large-scale prospective randomized trial of AZA plus VEN in the presence or absence of VK2 for AML patients with are unfit for intensive chemotherapy is warranted.

## Supporting information

**S1 Fig. Combination treatment of VK2 and AZA did not show synergistic cell death in AML cell lines.**
(PDF)

**S2 Fig. NOXA-knockdown attenuated enhanced cell death caused by coadministration of VK2 and VEN.**
(PDF)

**S3 Fig. Uncropped images of western blotting.**
(PDF)

**S1 File. Raw data used for graphs in Fig 2.**
(XLSX)

**S2 File. Raw data used for graphs in Fig 4.**
(XLSX)

**S3 File. Raw data used for graphs in Fig 5.**
(XLSX)

**S4 File. Raw data used for graphs in Fig 6.**
(XLSX)

**S5 File. Raw data used for graphs in S1 and S2 Figs.**
(XLSX)

## Author Contributions

**Conceptualization:** Tetsuzo Tauchi, Keisuke Miyazawa.

**Data curation:** Tetsuzo Tauchi, Shota Moriya, Seiichi Okabe, Hiromi Kazama, Naoharu Takano.

**Formal analysis:** Tetsuzo Tauchi, Naoharu Takano.

**Funding acquisition:** Shota Moriya, Naoharu Takano.

**Investigation:** Tetsuzo Tauchi, Shota Moriya, Seiichi Okabe, Hiromi Kazama, Naoharu Takano.

**Methodology:** Tetsuzo Tauchi, Shota Moriya, Seiichi Okabe, Keisuke Miyazawa, Naoharu Takano.

**Project administration:** Tetsuzo Tauchi, Keisuke Miyazawa, Naoharu Takano.

**Resources:** Tetsuzo Tauchi, Keisuke Miyazawa.

**Supervision:** Keisuke Miyazawa.

**Validation:** Tetsuzo Tauchi, Shota Moriya, Seiichi Okabe, Hiromi Kazama, Keisuke Miyazawa, Naoharu Takano.

**Visualization:** Tetsuzo Tauchi, Shota Moriya, Hiromi Kazama, Keisuke Miyazawa, Naoharu Takano.

**Writing – original draft:** Tetsuzo Tauchi, Shota Moriya, Keisuke Miyazawa, Naoharu Takano.

**Writing – review & editing:** Tetsuzo Tauchi, Shota Moriya, Keisuke Miyazawa, Naoharu Takano.

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
