## [Decision Letter · Decision Letter 0]

17 May 2024

PONE-D-24-12332Vitamin K2 sensitizes the efficacy of venetoclax in acute myeloid leukemia by targeting the NOXA-MCL-1 pathwayPLOS ONE

Dear Dr. Takano,

Thank you for submitting your manuscript to PLOS ONE. After careful consideration, we feel that it has merit but does not fully meet PLOS ONE’s publication criteria as it currently stands. Therefore, we invite you to submit a revised version of the manuscript that addresses the points raised during the review process.

We look forward to receiving your revised manuscript.

Kind regards,

Vivek Agrahari

Academic Editor

PLOS ONE

5. Please include captions for your Supporting Information files at the end of your manuscript, and update any in-text citations to match accordingly. Please see our Supporting Information guidelines for more information: http://journals.plos.org/plosone/s/supporting-information

Reviewers' comments:

Reviewer's Responses to Questions

**Comments to the Author**

1. Is the manuscript technically sound, and do the data support the conclusions?

Reviewer #1: Yes

Reviewer #2: Yes

2. Has the statistical analysis been performed appropriately and rigorously? 

Reviewer #1: Yes

Reviewer #2: Yes

3. Have the authors made all data underlying the findings in their manuscript fully available?

Reviewer #1: Yes

Reviewer #2: Yes

4. Is the manuscript presented in an intelligible fashion and written in standard English?

Reviewer #1: Yes

Reviewer #2: Yes

5. Review Comments to the Author

Reviewer #1: Hematologists are increasingly faced with the appearance of resistance to the combined azacitidine - venetoclax therapy administered to patients with AML. That is why this article is welcome and its results deserve to be popularized, even if the number of included patients is small. It is a research direction for the future, which must be encouraged.

The authors described the research methodology very clearly, presented the results well and adequately discussed the pathophysiological mechanisms involved, including in the light of data from recent literature.

It is good for the authors to also discuss the effects of vitamin K on normal myeloid and erythroid progenitors: promoting the differentiation of myeloid progenitors and an anti-apoptotic effect that seemed to be dominant in erythroid progenitors (Sada E, Abe Y, Ohba R, Tachikawa Y, Nagasawa E, Shiratsuchi M, Takayanagi R. Vitamin K2 modulates differentiation and apoptosis of both myeloid and erythroid lineages. Eur J Haematol. 2010 Dec;85(6):538-48. doi: 10.1111/j.1600-0609.2010.01530.x. PMID: 20887388).

The authors must emphasize the fact that, according to the latest research, venetoclax resistance of AML cells involves transcriptional and phenotypic heterogeneity (Mohanty V, Baran N, Huang Y, Ramage CL, Cooper LM, He S, Iqbal R, Daher M; CTD2 Research Network; Tyner JW, Mills GB, Konopleva M, Chen K. Transcriptional and phenotypic heterogeneity underpinning venetoclax resistance in AML. bioRxiv [Preprint]. 2024 Jan 30:2024.01.27.577579. doi: 10.1101/2024.01.27.577579. PMID: 38352538; PMCID: PMC10862759).

It follows that extensive research is needed to investigate different potential therapeutic targets and lead to therapeutic solutions, probably combined or personalized.

The expression in English is very good.

In my opinion, the article can be published after minimal corrections.

Reviewer #2: This is a well-written manuscript and one of its kind as a proof of concept to encourage large-scale clinical trials. However, there are some significant gaps, such as why AZA is not considered alongside VK2 in the cell-based studies. This manuscript requires minor revision, as listed below:

1. Why AZA in combination with vk2 was not studied?

2. NOXA-MCL pathway explanation should be included in the introduction? It is a major pathway of the study and clearly took place in the title?

3. Cell line name should be provided appropriately, like SKM-1 – full name

4. Cell viability and apoptosis were performed on the knock down cell line or conventional? It was not mentioned in the methods – line 142, 150, 157, 166. Use the appropriate cell name in the methods.

5. What is FSC and SSC? (forward scatter and side scatter? of flow cytometry?)

6. Mention cycle 1 again in the result (line 253)

7. The extremely high response rate of what? (line 246). Provide the meaningful sentence

8. Is this the first study showing VK2-induced ROS damage the mitochondria ? (line 282)

9. Provide the acronym at first place as possible, ELN, ECOP

6. PLOS authors have the option to publish the peer review history of their article (what does this mean?). If published, this will include your full peer review and any attached files.

Reviewer #1: No

Reviewer #2: No

---

## [Author Response · Author response to Decision Letter 0]

24 Jun 2024

POINT-BY-POINT RESPONSES TO THE REVIEWER’S COMMENTS

Thank you for your valuable comment. We have carefully revised our manuscript according to the reviewers’ advice.

Reviewer #1

Reviewer #1: 

Hematologists are increasingly faced with the appearance of resistance to the combined azacitidine - venetoclax therapy administered to patients with AML. That is why this article is welcome and its results deserve to be popularized, even if the number of included patients is small. It is a research direction for the future, which must be encouraged. The authors described the research methodology very clearly, presented the results well and adequately discussed the pathophysiological mechanisms involved, including in the light of data from recent literature.

Reply

We appreciate your positive comments on our paper and are grateful for your valuable comments. We have carefully revised the manuscript accordingly.

Reviewer #1-1: 

It is good for the authors to also discuss the effects of vitamin K on normal myeloid and erythroid progenitors: promoting the differentiation of myeloid progenitors and an anti-apoptotic effect that seemed to be dominant in erythroid progenitors (Sada E, Abe Y, Ohba R, Tachikawa Y, Nagasawa E, Shiratsuchi M, Takayanagi R. Vitamin K2 modulates differentiation and apoptosis of both myeloid and erythroid lineages. Eur J Haematol. 2010 Dec;85(6):538-48. doi: 10.1111/j.1600-0609.2010.01530.x. PMID: 20887388).

Reply

Thank you for your comment. According to your suggestions, we mentioned the effects of vitamin K2 on normal myeloid cells in the “Introduction” section as follows: The effect of VK2 on normal hematopoietic progenitors also verified that VK2 induces normal myeloid progenitor differentiation and exhibits an anti-apoptotic effect on normal erythroid progenitors. (Page 5, Line 79-81)

Reviewer #1-2: 

The authors must emphasize the fact that, according to the latest research, venetoclax resistance of AML cells involves transcriptional and phenotypic heterogeneity (Mohanty V, Baran N, Huang Y, Ramage CL, Cooper LM, He S, Iqbal R, Daher M; CTD2 Research Network; Tyner JW, Mills GB, Konopleva M, Chen K. Transcriptional and phenotypic heterogeneity underpinning venetoclax resistance in AML. bioRxiv [Preprint]. 2024 Jan 30:2024.01.27.577579. doi: 10.1101/2024.01.27.577579. PMID: 38352538; PMCID: PMC10862759).

Reply:

Thank you for your comment. As the reviewer pointed out, we have added the following explanation:

Recent research on transcriptional and phenotypic heterogeneity within or among patients has shown that resistance to venetoclax in AML is not solely dependent on MCL1 expression [44]. Further studies is needed to identify the type of AML sensitized by VK2 in combination with venetoclax. (Page 25, Line 452-455)

It follows that extensive research is needed to investigate different potential therapeutic targets and lead to therapeutic solutions, probably combined or personalized.

The expression in English is very good.

In my opinion, the article can be published after minimal corrections.

 

Reviewer #2

Reviewer #2: This is a well-written manuscript and one of its kind as a proof of concept to encourage large-scale clinical trials. However, there are some significant gaps, such as why AZA is not considered alongside VK2 in the cell-based studies. This manuscript requires minor revision, as listed below:

Reply:

We appreciate your favorable appraisal of our paper and are grateful for your valuable comments. We performed the additional experiment suggested and have carefully revised our manuscript accordingly.

Reviewer #2-1:

Why AZA in combination with vk2 was not studied?

Reply:

Thank you for your comment. As we have explained in the Result section, “Because all patients simultaneously received a specific BCL-2 inhibitor, VEN, with VK2 (45 mg/day, po) during the entire cycle of chemotherapy, we hypothesized that the drug combination of VEN plus VK2 rather than AZA plus VK2 might attribute to the high response rate.”, we administered VEN and VK2 during the entire cycle, but administered AZA on Day 1–7. Therefore, we focused on the combined effects of VEN and VK2. To test this hypothesis, we assessed whether the combination of AZA and VK2 synergistically induces cell death. However, this assay did not show any synergistic effects. Therefore, we added the following supporting figure and sentences: 

“In addition, the combination of AZA and VK2 did not show synergistic effects in contrast to the combination of VEN and VK2 (Fig S1).” (Page 17, Line 291-292)

Reviewer #2-2:

NOXA-MCL pathway explanation should be included in the introduction? It is a major pathway of the study and clearly took place in the title?

Reply:

Thank you for your comment. As suggested, we have explained the NOXA–MCL-1 pathway in the “Introduction” section as below.

For example, the anti-apoptotic protein MCL-1 is upregulated in VEN-resistant AML [6]. (Page 3, Line 55-56)

NOXA is a pro-apoptotic BH3-only protein that interacts with and suppresses MCL-1. (Page 5, Line 89-90)

Reviewer #2-3:

Cell line name should be provided appropriately, like SKM-1 – full name

Reply

Thank you for your comment. According to to your suggestion, we corrected the cell line name U937 to “U-937” in the text and figures.

(Page 7 Line 120, Page 9 Line 154, Page 17 Line 296, Figure 2D.)

Reviewer #2-4:

Cell viability and apoptosis were performed on the knock down cell line or conventional? It was not mentioned in the methods – line 142, 150, 157, 166. Use the appropriate cell name in the methods.

Reply

Thank you for your comment. As suggested, we have added the cell line names to the methods as below:

“Twenty-four hours after the transfection, the cells were treated with VK2, VEN, or VK2 plus VEN for 48 h and used for cell viability assay or immunoblotting as described below.” (Page 8, Line 147-149)

“with HL-60 (wild-type and NOXA-KO), SKM-1 (wild-type and NOXA-KO), THP-1, U-937, and MOLM-14 cell lines (Page 9, Line 153-154)

“HL-60 and SKM-1 cells were treated with VK2 with or without VEN for 48 h and resuspended in Annexin V-binding buffer at 1×106 cells/ml” (Page 9, Line 161-162)

“HL-60 and SKM-1 cells were treated with either VK2, VEN, or VK2 plus VEN for 48 h and then” (Page 9, Line 168)

“HL-60 (wild-type or NOXA-KO) and SKM-1 (wild-type or NOXA-KO) cells were treated with VK2, VEN, or VK2 plus VEN for 24 or 48 h. Then” (Page 10, Line 178-179)

Reviewer #2-5:

What is FSC and SSC? (forward scatter and side scatter? of flow cytometry?)

Reply

Thank you for your comment. We apologize for the lack of explanation and have added the explanation as below:

“Healthy cells gated with forward scatter (FSC) and side scatter (SSC) of flow cytometry were used for the analysis.” (Page 12, Line 220-221)

Reviewer #2-6:

Mention cycle 1 again in the result (line 253

Reply:

Thank you for your comment. We mentioned “cycle 1” again in the result section as below:

“Because all patients have simultaneously received a specific BCL-2 inhibitor, VEN, with VK2 (45 mg/day, po) during the entire cycle of chemotherapy, we hypothesized that the drug combination of VEN plus VK2 rather than AZA plus VK2 might be attributed to the higher response rate after cycle 1.” (Page 17, Line 285)

Reviewer #2-7:

The extremely high response rate of what? (line 246). Provide the meaningful sentence

Reply:

Thank you for the comment. We have revised the explanation as below.

 “The extremely high CR/CRi rate” (Page 16, Line 279)

Reviewer #2-8:

Is this the first study showing VK2-induced ROS damage the mitochondria ? (line 282)

Reply:

Thank you for your comment. Duan et al. (2016) have reported that ROS scavenger recovers VK2-induced mitochondrial membrane potential decrement. (PMID: 27570977)

Therefore we added the new sentence as below.

Similar results have also been observed in bladder cancer cells [30]. (Page 20, Line 341)

Reviewer #2-9:

Provide the acronym at first place as possible, ELN, ECOP

Reply:

Thank you for your comment. As suggested, we have defined the acronyms ELN and ECOP as follows:

“Most patients had high-risk features: 13 patients (68.4%) had ELN (European Leukemia Network) adverse risk, 47% ECOG PS (Eastern Cooperative Oncology Group Performance Status) 2 >, and 2 (10.5%) had TP53 deletion.” (Page 14, Line 248-249)

“ECOG PS, Eastern Cooperative Oncology Group Performance Status” (Page 16, Line 263 Table 1 legend)

---

## [Decision Letter · Decision Letter 1]

10 Jul 2024

Vitamin K2 sensitizes the efficacy of venetoclax in acute myeloid leukemia by targeting the NOXA-MCL-1 pathway

PONE-D-24-12332R1

Dear Dr. Takano,

We’re pleased to inform you that your manuscript has been judged scientifically suitable for publication and will be formally accepted for publication once it meets all outstanding technical requirements.

Kind regards,

Vivek Agrahari

Academic Editor

PLOS ONE

Additional Editor Comments (optional):

Reviewers' comments:

Reviewer's Responses to Questions

**Comments to the Author**

1. If the authors have adequately addressed your comments raised in a previous round of review and you feel that this manuscript is now acceptable for publication, you may indicate that here to bypass the “Comments to the Author” section, enter your conflict of interest statement in the “Confidential to Editor” section, and submit your "Accept" recommendation.

Reviewer #2: All comments have been addressed

2. Is the manuscript technically sound, and do the data support the conclusions?

Reviewer #2: Yes

3. Has the statistical analysis been performed appropriately and rigorously? 

Reviewer #2: Yes

4. Have the authors made all data underlying the findings in their manuscript fully available?

Reviewer #2: Yes

5. Is the manuscript presented in an intelligible fashion and written in standard English?

Reviewer #2: Yes

6. Review Comments to the Author

Reviewer #2: (No Response)

7. PLOS authors have the option to publish the peer review history of their article (what does this mean?). If published, this will include your full peer review and any attached files.

Reviewer #2: No

---

## [Editor Report · Acceptance letter]

15 Jul 2024

PONE-D-24-12332R1 

PLOS ONE

Dear Dr. Takano, 

I'm pleased to inform you that your manuscript has been deemed suitable for publication in PLOS ONE. Congratulations! Your manuscript is now being handed over to our production team.

Kind regards, 

on behalf of

Dr. Vivek Agrahari 

Academic Editor

PLOS ONE